# Toward Robust In-Context Learning: Leveraging Out-of-distribution Proxies for Target Inaccessible Demonstration Retrieval

## Abstract

Although studies have demonstrated that Large Language Models (LLMs) can perform well on Out-of-Distribution (OOD) tasks, their advantage tends to diminish as the distribution shift becomes more severe. Consequently, researchers aim to retrieve distributionally similar and informative demonstrations from the available source domain to boost the inference capabilities of LLMs. However, in practical scenarios where the target domain is inaccessible, evaluating the unknown distribution is challenging, which indirectly impacts the quality of the selected demonstrations. To address this problem, we propose **DOPA**, a demonstration search framework that incorporates an OOD proxy to approximate the inaccessible target domain and guide the retrieval process. Building on proxy-based evaluation, DOPA further introduces a Mahalanobis distance-based global diversity constraint to ensure sufficient diversity among the retrieved demonstrations. Experimental results on multiple LLMs and natural language understanding tasks demonstrate that DOPA effectively enhances robustness in OOD settings[1].

## 1 Introduction

Large language models (LLMs) have played an indispensable role in the field of natural language processing (NLP), achieving remarkable performance across a wide range of tasks Chang et al. (2024); Song et al. (2025). Among various prompting strategies, in-context learning (ICL) has emerged as one of the most widely adopted approaches, wherein providing a few-shot demonstration can effectively guide the model toward improved reasoning and prediction Min et al. (2022). However, recent studies have revealed that the performance of LLMs can degrade significantly in out-of-distribution (OOD) scenarios Yuan et al. (2023); Wang et al. (2025), where the demonstrations exhibit substantial distributional differences from the target domain. This has motivated researchers to explore various methods for obtaining more effective demonstrations.

Retrieval Luo et al. (2024) and augmentation Shu et al. (2024) are two commonly used approaches for obtaining effective samples. The former searches for the most relevant examples within a specific domain, while the latter rewrites existing samples to reduce their discrepancy with the target instance. Demonstration retrieval relies on a retriever. Some off-the-shelf metrics, such as Bm25 Agrawal et al. (2023), sentence encoder-based similarity Liu et al. (2022), model influence Peng et al. (2024); S. et al. (2024), and misconfidence Xu & Zhang (2024), can support general-purpose retrieval strategies. While other approaches aim to train a dense retriever to obtain more task-relevant retrieval results Cheng et al. (2023); Li et al. (2023a). Augmentation, on the other hand, focuses on adapting existing samples to better match the distributional characteristics of the target instance O'Brien et al. (2024); Madine (2024). ***However, in real-world applications, inaccessible target domain hinders the ability to obtain domain-aligned demonstrations, often resulting in degraded performance Song et al. (2024a).***

To address the aforementioned challenge, we propose a **d**emonstration **o**ptimization framework based on OOD **p**roxy **a**ssessment (termed **DOPA**). This framework quantifies the utility of source-domain samples in the absence of target-domain access, and leverages the quantification results to

---

[1]https://anonymous.4open.science/r/ood_code

guide demonstration retrieval. At its core, DOPA introduces an OOD proxy as a principled approximation to the unknown target distribution Zhang & Wischik (2022), which is composed of two components: a source proxy and a target proxy. The source proxy is defined as an instruction-tuned LLM trained on the source domain to fully adapt to the source distribution, while the target proxy corresponds to the original, unmodified version of the same LLM. The perplexity ratio between their predictions on identical input samples is adopted as the OOD score for those samples Nalisnick et al. (2019). This OOD score serves to estimate the degree of familiarity of source-domain samples with the target domain in the absence of target-domain information. It is further integrated with representational similarity to predictable samples for candidate selection. The validity of the OOD score is theoretically supported through a bounded proxy error analysis. Moreover, to enhance the diversity of retrieved demonstrations, we incorporate a Mahalanobis distance-based search strategy into the retrieval process. By relying on the OOD proxy, DOPA is capable of identifying informative demonstrations solely within the source domain, without requiring any samples from the target domain. Extensive experiments show that DOPA consistently outperforms baseline approaches across diverse LLMs and natural language understanding tasks. In addition, we provide a multi-dimensional analysis that demonstrates the effectiveness of the proxy in selecting samples that exhibit behavioral similarity to those in the target domain. Our contributions are as follows:

(i) We propose a method that leverages OOD proxies to extract distribution-aligned samples, and we theoretically demonstrate the soundness of the proxy through a bounded proxy error guarantee. (ii) We propose a target-agnostic demonstration retrieval framework based on OOD proxies, which combines proxy results and contextual diversity to enhance the quality of demonstration selection. (iii) Experimental results on multiple NLP tasks and across various LLMs demonstrate that the proposed method effectively enhances OOD robustness in ICL.

## 2 RELATED WORK

**Demonstration Retrieval**. Despite the impressive performance demonstrated by ICL, an increasing number of studies have shown its sensitivity to the choice of demonstrations Song et al. (2024b). To obtain more effective demonstrations, a natural idea is to search over candidate samples within a constrained space Luo et al. (2024). Depending on whether the retrieval tool has been trained, demonstration search can be divided into off-the-shelf retrieval and retrieval based on fine-tuned models. Term-based similarity has been widely used for demonstration retrieval, with BM25 being one of the most popular scoring metrics Agrawal et al. (2023); Ye et al. (2023). In addition, several sentence embedding models, such as SBERT Wang et al. (2024), RoBERTa Liu et al. (2022), and SimCSE Gao et al. (2021), have also been widely used to compute inter-sample similarity and optimize demonstration selection. Moreover, some approaches assess the influence of individual samples on model predictions to select high-impact examples for demonstrations Peng et al. (2024); S. et al. (2024). Off-the-shelf retrieval methods may yield suboptimal results, as they do not incorporate task-specific information. Therefore, some methods have explored leveraging feedback signals from LLMs to distinguish between important and unimportant samples, and further optimize the retriever for specific tasks using objectives such as ranking Li et al. (2023a), contrastive learning Cheng et al. (2023); Luo et al. (2023), and diversity Ye et al. (2023). But these methods often rely on feedback from LLMs, which leads to higher computational complexity.

**OOD Robustness in ICL**. In ICL settings, distribution shifts can lead to significant performance drops, revealing the models' sensitivity and lack of robustness to unseen or mismatched domains Yuan et al. (2023); Wang et al. (2025). The presence of a distributional gap may render demonstration retrieval strategies ineffective, as the retrieved examples may no longer align with the target task semantics. Therefore, some approaches have further explored the effectiveness of demonstration augmentation. Some approaches have introduced external knowledge, such as linguistic rules Jiang et al. (2024) or human feedback Bai et al. (2024), to fine-tune LLMs. Nevertheless, some studies have questioned the necessity of fine-tuning, arguing that LLMs may already possess the inherent capability to handle OOD data effectively Uppaal et al. (2023); Zhang et al. (2024). As a result, semantic rewriting has been introduced to prompt LLM to revise a given source sample, aiming to better align it with the target domain O'Brien et al. (2024); Madine (2024).

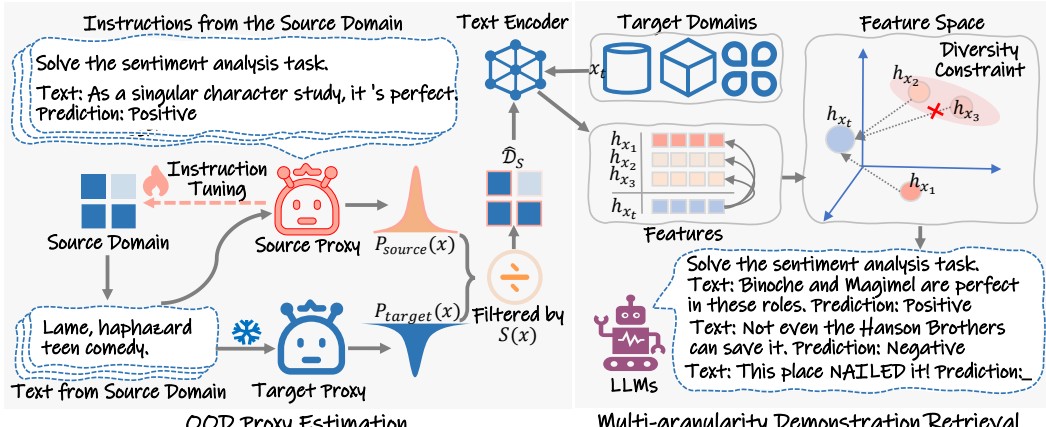

Figure 1: The model architecture of DOPA based on the sentiment analysis task. First, DOPA performs task-specific instruction tuning on the source domain to obtain a source proxy based on any given LLM. Correspondingly, an identical LLM without fine-tuning, which preserves the prior knowledge of the target domain, is employed as the target-domain proxy. For the same input, the ratio between the two proxies is employed as an OOD proxy estimation, which is further combined with similarity and diversity to support multi-granularity demonstration search.

## 3 METHOD

### 3.1 TASK DEFINITIONS AND MODEL DESCRIPTION

Our model description begins with some definitions. In the OOD setting, LLMs $\mathcal{M}$ are restricted to using data from $\mathcal{D}_S$ to perform ICL, and are expected to make predictions on any sample $x_t$ from $\mathcal{D}_T$ as accurately as possible. During the inference process of LLMs, all samples from $\mathcal{D}_T$ other than $x_t$ are strictly inaccessible, preventing the model from making decisions by referencing samples from a similar distribution. For ICL, a prompt $\mathcal{P}x_t$ is constructed by selecting $N \times |Y|$ labeled examples $(x^{(j)}, y^{(j)})_{j=1}^{N \times |Y|}$ from $\mathcal{D}_S$, which are then concatenated with $x_t$ and fed into any $\mathcal{M}$. Here, $|Y|$ denotes the size of the label space. Then, the LLM produces a prediction $\hat{y}_t = \mathcal{M}(\mathcal{P}x_t)$. In different task settings, $\hat{y}_t$ can take various forms depending on the output space. For classification tasks, it typically corresponds to a token representing a label category (e.g., positive or negative), while for generative tasks, it may be a string representing the desired output.

As illustrated in Figure 1, DOPA comprises two main components: OOD proxy estimation and multi-granularity demonstration retrieval. The proxy estimation module assesses the proximity of source domain samples to the target domain using an OOD proxy, while the demonstration retrieval module selects appropriate examples by jointly optimizing semantic similarity and diversity constraints. The retrieved demonstrations are then used for ICL.

### 3.2 OOD PROXY ESTIMATION

The goal of the OOD proxy estimation is to evaluate the utility of source domain samples to select those that are more aligned with the target domain. But without access to the target domain, it is difficult to accurately assess the target distribution. Therefore, inspired by prior work on OOD detection Ren et al. (2019); Zhang & Wischik (2022), we construct an OOD proxy to approximate the target domain distribution, and compute the OOD score of any sample via the proxy. The OOD score is then used to guide sample selection from $\mathcal{D}_S$.

**Proxy Construction**. The OOD proxy consists of two components: the source proxy and the target proxy, which ideally model the source and target distributions, respectively. For the former, an intuitive approach is to instruction-tune LLMs on the source domain so that the model can better adapt to the source distribution. In DOPA, the instructions for source proxy are encapsulated in the same format as in ICL, aiming to prompt LLMs to produce reasonable task-related predictions. As

for the target domain, since the target domain distribution is unknown, some methods propose a general approach by replacing the target-domain proxy with a uniform distribution Bishop (1993); Nalisnick et al. (2019). Such a strong assumption is inherently destined to yield suboptimal results as shown in Lemma 1. Given that LLMs are pretrained on extensive corpora, it is reasonable to assume that they implicitly encode a broad spectrum of linguistic and factual knowledge. As such, LLMs can act as weak proxies for the target distribution, particularly in few-shot settings Zhang et al. (2024).

**Sample Screening based on OOD Score**. Given the aforementioned proxies, DOPA further assumes that if a sample exhibits divergent behavior under these two proxies, it may suggest an inherent bias or a stronger alignment toward one specific domain. This facilitates domain discrimination in the absence of any auxiliary target domain samples. In the previous research, the likelihood ratio is one of the most commonly used detection criteria for the divergent behavior Ren et al. (2019); Zhang & Wischik (2022):

$$S(x) = \frac{P_{target}(x)}{P_{source}(x)} \approx \underbrace{\frac{P_{target}^{proxy}(x)}{P_{source}^{proxy}(x)}}_{\text{OOD proxy}}, \tag{1}$$

where $P_{source}(x)$ and $P_{target}(x)$ represent the behavior of models with the source and target domain distributions when given the same input sample $x$, respectively. Under distributional uncertainty, the OOD proxy is used as their approximation. To support the validity of OOD proxy estimation, we further establish a theoretical guarantee on the boundedness of proxy error under mild assumptions.

**Theorem 1** (Proxy Error Bound). *Let $P_{\text{target}}$ and $P_{\text{source}}$ be the true probability distributions of the target and source domain, let $P_{\text{target}}^{\text{proxy}}$ and $P_{\text{source}}^{\text{proxy}}$ be the corresponding proxy distributions. Suppose there exist constants $\varepsilon_t \geq 0$, $\varepsilon_s \geq 0$, $m_t > 0$, and $m_s > 0$ such that the following hold:*

- *The Kullback-Leibler divergences are bounded: $D_{\text{KL}}(P_{\text{target}} \parallel P_{\text{target}}^{\text{proxy}}) \leq \varepsilon_t, D_{\text{KL}}(P_{\text{source}} \parallel P_{\text{source}}^{\text{proxy}}) \leq \varepsilon_s$. The proxy distributions have pointwise lower bounds: $P_{\text{target}}(x), P_{\text{target}}^{\text{proxy}}(x) \geq m_t, \quad P_{\text{source}}(x), P_{\text{source}}^{\text{proxy}}(x) \geq m_s$.*

*Then, for all $x$, the error in the log-likelihood ratio satisfies:*

$$\left| \log \frac{P_{\text{target}}(x)}{P_{\text{source}}(x)} - \log \frac{P_{\text{target}}^{\text{proxy}}(x)}{P_{\text{source}}^{\text{proxy}}(x)} \right| \leq \frac{\varepsilon_t}{m_t} + \frac{\varepsilon_s}{m_s}.$$

**Lemma 1** (Error Bound with Uniform Proxy). *Building upon Theorem 1, if a uniform distribution is used as the proxy for the target domain, it is easier to result in a looser upper bound on the error.*

Due to space limitations, the proof of the above theorem is provided in Appendix B. Such a uniform bound certifies that the proxy-based score deviates from the true likelihood ratio by at most a known quantity, thereby providing theoretical assurance for reliable estimation. In the case of autoregressive LLMs, perplexity Wuhrmann et al. (2025) is commonly employed to quantify the model's familiarity with a given text $x$: $PPL(x) = exp\left(-\frac{1}{m} \sum_{i=1}^{m} logP(w_i|w_{<i})\right)$, where $m$ is the total number of tokens in $x$, and $P(w_i|w_{<i})$ is the conditional probability of the language model predicting the $i$-th token. Therefore, to conform to the $log$ form as stated in Theorem 1, we adopt the log-perplexity difference as a more numerically stable alternative:

$$S(x) = logPPL_{target}^{proxy}(x) - logPPL_{source}^{proxy}(x). \tag{2}$$

Ideally, if the value of $S(x)$ is relatively low, it exhibits higher perplexity under the source-domain proxy and lower perplexity under the target-domain proxy, which further indicates that the sample is more aligned with the target domain and should therefore be prioritized for constructing demonstrations. By performing a single pass over $\mathcal{D}_S$, we can obtain a potential subset $\hat{\mathcal{D}}_S$ that is closer to the target domain distribution by selecting the $k$ samples with the lowest OOD scores.

### 3.3 DEMONSTRATION RETRIEVAL

Although the OOD scores help identify source domain samples that are more likely to align with the target domain, the resulting coarse-grained subset still requires further refinement to construct effective demonstrations. Existing studies have provided strong support for the demonstration search

---

**Algorithm 1** Demonstration Retrieval Process of DOPA

---

**Input**: Proxy-filtered set $\hat{\mathcal{D}}_S$, test sample $x_t$.
**Parameter**: Demonstration quantity $N \times |Y|$, initialized candidate set $C$.
**Output**: Final demonstration set $\mathcal{D}_{demo}$ with size $N$.

1: Init $\mathcal{D}_{demo}$ by Eq.3, sort $\hat{\mathcal{D}}_S$ in ascending order according to $sim(h_{x_i}, h_t)$, counter$\leftarrow$ 0.
2: **while** $|\mathcal{D}_{demo}| < N \times |Y|$ **do**
3:    $\hat{x} \leftarrow \hat{\mathcal{D}}_S[C + counter]$.
4:    **if** $Div_{\mathcal{D}_{demo}} \leq Div_{\{\hat{x}\} \cup \mathcal{D}_{demo}}$ **then**
5:       $\mathcal{D}_{demo} \leftarrow \{\hat{x}\} \cup \mathcal{D}_{demo}$.
6:    **end if**
7:    counter$\leftarrow$counter + 1.
8: **end while**
9: **return** $\mathcal{D}_{demo}$

---

process Liu et al. (2022); Agrawal et al. (2023), a general approach is to adopt an off-the-shelf text representation model to encode candidate texts into vectors and rank the most relevant demonstrations based on their cosine similarity with the test sample. But one limitation of proxy-based OOD scoring lies in its reliance on language model perplexity, which primarily captures token-level fluency and distributional similarity. As a result, it may implicitly favor shorter texts or those conforming to high-frequency linguistic patterns Holtzman et al. (2020), leading to reduced diversity in the selected sample pool and potentially impairing the quality of the retrieved demonstrations. To address this issue, we further introduce a global diversity constraint to improve the overall quality of the retrieved demonstrations. Specifically, for each sample representation $h_{x_i}$ corresponding to the proxy-filtered set $\hat{\mathcal{D}}_S$, we initialize a candidate sample set $\mathcal{D}_{demo}$ based on the similarity of the representations to $h_t$:

$$\mathcal{D}_{demo} = arg \max_C \{sim(h_{x_i}, h_t)\}_{i=1}^{|\hat{\mathcal{D}}_S|}, \tag{3}$$

where $C$ is the number of samples in the initialized candidates[2]. Subsequently, the mean pairwise Mahalanobis distance Li et al. (2023b) among samples in $\mathcal{D}_{demo}$ is used to quantify the diversity:

$$Div = \frac{2}{|\mathcal{D}_{demo}|(|\mathcal{D}_{demo}| - 1)} \sum_{i<j} \sqrt{\mathbf{D}_{ij}^{\top} \Sigma^{-1} \mathbf{D}_{ij}}, \tag{4}$$

where $\mathbf{D}_{ij} = h_{x_i} - h_{x_j}$, $\Sigma$ is the empirical covariance matrix computed over all samples. The Mahalanobis distance is adopted because it accounts for the correlations between samples while measuring diversity, which helps impose constraints on similarity-based retrieval results. If a new sample $\hat{x} \in \{\hat{\mathcal{D}}_S - \mathcal{D}_{demo}\}$ does not lead to a decrease in overall diversity i.e. $Div_{\mathcal{D}_{demo}} \leq Div_{\{\hat{x}\} \cup \mathcal{D}_{demo}}$, it is retained. This process continues until the number of samples meets the required threshold for constructing demonstrations. The final selected demonstration set is used for ICL. The above procedure is summarized in Algorithm 1. After obtaining sufficient samples, we construct demonstrations in a fixed label order to prevent bias introduced by orders, and use them for ICL.

## 4 EXPERIMENT

### 4.1 EXPERIMENTAL SETUP

We conduct experiments on the OOD-specific benchmark BOSS Yuan et al. (2023), which includes three classification tasks, Sentiment Analysis (SA), Toxicity Detection (TD), and Natural Language Inference (NLI) as well as one generation task, Named Entity Recognition (NER). All instruction templates follow the format provided in the original BOSS paper. In addition, we compare our proposed method DOPA with various baseline approaches to comprehensively demonstrate its advantages. These baselines include: **Random** Peng et al. (2024), **KNN** Liu et al. (2022), **DrICL** Luo et al. (2023), **Rewrite** Madine (2024), and **InfICL** S. et al. (2024), where Rewrite refers to the data augmentation-based method, while the others are demonstration retrieval-based methods. All

---

[2]We specify the value of $C$ in the detailed experimental settings.

| LLMs | Methods | SA | | | | TD | | | | NLI | | | |
|---|---|---|---|---|---|---|---|---|---|---|---|---|---|
| | | dynasent | semeval | sst | **avg** | implicit | adv | toxigen | **avg** | wanli | anli | cnli | **avg** |
| GPT2-xl | Random | 36.33 | 49.28 | 47.70 | 44.44 | 50.47 | 50.20 | 50.60 | 50.42 | 34.23 | 32.23 | 39.22 | 35.23 |
| | KNN | 35.89 | 45.92 | 51.17 | 44.33 | 47.67 | 47.50 | 48.50 | 47.89 | 33.57 | 33.50 | 45.05 | 37.37 |
| | DrICL | 37.00 | 47.66 | 52.76 | 45.81 | 49.70 | 51.38 | 48.23 | 49.77 | 32.03 | 33.33 | 47.20 | 37.52 |
| | Rewrite | 36.00 | 45.84 | 50.61 | 44.15 | 46.90 | 45.72 | 49.37 | 47.33 | 34.00 | 32.77 | 45.38 | 37.38 |
| | InfICL | 36.61 | 49.26 | 46.95 | 44.27 | 49.90 | 50.33 | 50.17 | 50.13 | 33.80 | 32.40 | 24.25 | 30.15 |
| | DOPA* | 38.23 | 48.44 | 59.61 | **48.76** | 51.67 | 53.29 | 50.50 | **51.82** | 34.93 | 33.43 | 45.77 | **38.04** |
| LLaMA3.2-3B | Random | 53.81 | 47.86 | 66.26 | 55.98 | 57.70 | 55.20 | 65.70 | 59.53 | 37.70 | 35.50 | 42.75 | 38.65 |
| | KNN | 52.63 | 45.76 | 65.42 | 54.60 | 56.63 | 53.29 | 51.03 | 53.65 | 37.20 | 34.67 | 44.24 | 38.70 |
| | DrICL | 56.05 | 46.08 | 67.10 | 56.41 | 57.83 | 56.18 | 64.80 | 59.61 | 36.50 | 33.87 | 42.95 | 37.77 |
| | Rewrite | 53.92 | 45.06 | 64.29 | 54.43 | 51.57 | 57.04 | 62.70 | 57.10 | 36.43 | 35.60 | 42.75 | 38.26 |
| | InfICL | 53.35 | 46.80 | 64.39 | 54.84 | 56.33 | 55.20 | 65.57 | 59.03 | 36.23 | 36.00 | 40.65 | 37.63 |
| | DOPA* | 55.71 | 53.28 | 68.88 | **59.29** | 57.87 | 56.45 | 65.30 | **59.87** | 38.40 | 35.87 | 43.19 | **39.15** |
| Gemma2-2B | Random | 56.47 | 47.06 | 66.45 | 56.66 | 55.57 | 56.51 | 63.93 | 58.67 | 33.37 | 33.00 | 42.66 | 36.34 |
| | KNN | 55.29 | 47.28 | 66.26 | 56.28 | 53.20 | 47.89 | 63.87 | 54.99 | 33.50 | 32.93 | 41.61 | 36.01 |
| | DrICL | 57.67 | 47.20 | 67.10 | 57.32 | 55.43 | 56.45 | 61.17 | 57.68 | 33.73 | 33.57 | 45.43 | 37.58 |
| | Rewrite | 57.91 | 47.12 | 67.01 | 57.35 | 48.57 | 51.84 | 61.84 | 54.08 | 33.70 | 33.33 | 45.29 | 37.44 |
| | InfICL | 58.07 | 45.38 | 64.57 | 56.01 | 55.63 | 57.50 | 59.90 | 57.68 | 33.27 | 32.93 | 45.29 | 37.16 |
| | DOPA* | 57.24 | 47.70 | 68.13 | **57.69** | 56.53 | 58.50 | 65.73 | **60.12** | 33.37 | 33.07 | 46.10 | 37.51 |
| Qwen3-1.7B | Random | 62.82 | 60.90 | 69.17 | 64.29 | 54.97 | 52.89 | 67.20 | 58.35 | 41.30 | 35.17 | 39.12 | 38.53 |
| | KNN | 60.75 | 58.32 | 70.67 | 63.25 | 56.10 | 50.13 | 65.73 | 57.32 | 41.77 | 35.20 | 37.83 | 38.27 |
| | DrICL | 61.54 | 60.16 | 70.38 | 64.03 | 54.07 | 55.86 | 66.37 | 58.76 | 42.77 | 35.33 | 37.49 | 38.53 |
| | Rewrite | 60.38 | 54.72 | 70.29 | 61.80 | 51.33 | 57.50 | 61.97 | 56.93 | 39.47 | 36.63 | 38.79 | 38.30 |
| | InfICL | 62.10 | 61.12 | 69.92 | 64.38 | 55.30 | 56.83 | 65.23 | 59.12 | 40.80 | 35.57 | 40.03 | 38.80 |
| | DOPA* | 63.35 | 59.64 | 71.79 | **64.93** | 55.47 | 56.45 | 65.73 | **59.22** | 42.37 | 36.47 | 40.94 | **39.93** |
| GPT4o-mini | KNN | 67.67 | 60.50 | 78.17 | 68.78 | 57.63 | 57.13 | 82.00 | 65.58 | 37.67 | 39.67 | 26.83 | 34.72 |
| | InfICL | 63.83 | 54.17 | 80.67 | 66.22 | 59.38 | 62.88 | 84.15 | 68.79 | 38.38 | 38.00 | 32.83 | 36.39 |
| | DOPA* | 67.83 | 62.00 | 81.17 | **70.33** | 59.50 | 65.25 | 83.25 | **69.33** | 38.67 | 40.83 | 32.67 | **37.39** |
| GPT3.5-turbo | KNN | 67.17 | 60.00 | 78.83 | 68.67 | 58.38 | 60.50 | 82.23 | 67.04 | 38.00 | 40.00 | 30.83 | 36.28 |
| | InfICL | 66.00 | 55.83 | 80.17 | 67.33 | 60.00 | 64.13 | 84.50 | **69.54** | 37.67 | 39.50 | 32.50 | **36.56** |
| | DOPA* | 68.00 | 61.17 | 80.50 | **69.89** | 58.50 | 66.13 | 82.63 | 69.08 | 38.17 | 39.33 | 32.00 | 36.50 |

Table 1: The performance (accuracy %) on classification tasks, * indicates that the results based on the LLM among all the datasets are statistically significant under the Wilcoxon Signed-Rank Test ($p \leq 0.05$). The calculation process of significance is presented in Appendix C.3.

| LLMs | Methods | NER | | | LLMs | Methods | NER | | |
|---|---|---|---|---|---|---|---|---|---|
| | | wnut | ener | **avg** | | | wnut | ener | **avg** |
| GPT2-xl | Random | 22.85 | 32.83 | 27.84 | Qwen3-1.7B | Random | 25.92 | 28.71 | 27.32 |
| | KNN | 51.03 | 54.47 | 52.75 | | KNN | 38.74 | 50.62 | 44.68 |
| | DrICL | 19.65 | 23.29 | 21.47 | | DrICL | 39.59 | 45.33 | 42.46 |
| | DOPA | 51.32 | 56.82 | **54.07** | | DOPA | 41.25 | 49.62 | **45.44** |
| LLaMA3.2-3B | Random | 29.54 | 41.13 | 35.34 | GPT4o-mini | Random | 42.98 | 25.20 | 34.09 |
| | KNN | 33.50 | 40.97 | 37.23 | | KNN | 44.13 | 24.02 | 34.08 |
| | DrICL | 28.87 | 32.08 | 30.47 | | DrICL | 46.72 | 25.50 | 36.11 |
| | DOPA | 37.19 | 41.40 | **39.29** | | DOPA | 51.49 | 38.19 | **44.84** |
| Gemma2-2B | Random | 20.70 | 22.88 | 21.79 | GPT3.5-turbo | Random | 44.84 | 28.97 | 36.90 |
| | KNN | 29.64 | 37.70 | **33.67** | | KNN | 46.65 | 25.11 | 35.88 |
| | DrICL | 24.30 | 27.12 | 25.71 | | DrICL | 45.30 | 28.23 | 36.77 |
| | DOPA | 29.67 | 37.37 | 33.52 | | DOPA | 50.00 | 35.29 | **42.65** |

Table 2: The performance on NER tasks.

the methods are implemented in different LLMs to verify the adaptability of the proposed methods, including GPT2-xl[3], Qwen3-1.7B[4], Gemma2-2B[5], and LLaMA3.2-3B[6]. In addition, to investigate the performance of the proposed method on closed-source models, we also conduct experiments on GPT4o-mini and GPT3.5-turbo, and compare them with KNN, InfICL, and Rewrite. Additional experimental settings can be found in the Appendix C.

## 4.2 EXPERIMENTAL RESULTS

We present the comparison results of DOPA with the aforementioned baseline methods on different LLMs in Table 1 and Table 2. We do not compare InfICL and Rewrite on the NER task because, for token-level tasks, the influence of individual samples is difficult to quantify, and sentence rewriting

---

[3]https://huggingface.co/openai-community/gpt2-xl

[4]https://huggingface.co/Qwen/Qwen3-1.7B

[5]https://huggingface.co/google/gemma-2b

[6]https://huggingface.co/meta-llama/Llama-3.2-3B

| Variants | LLaMA3.2-3B | | | | | Qwen3-1.7B | | | | |
|---|---|---|---|---|---|---|---|---|---|---|
| | **SA** | **TD** | **NLI** | **NER** | **avg** | **SA** | **TD** | **NLI** | **NER** | **avg** |
| DOPA$_{-mah}$ | 56.59 | 58.67 | 38.66 | 37.13 | 47.76$_{\downarrow 1.64}$ | 64.33 | 58.54 | 38.92 | 44.26 | 51.51$_{\downarrow 0.87}$ |
| DOPA$_{-sim}$ | 56.15 | 57.62 | 37.24 | 36.24 | 46.81$_{\downarrow 2.59}$ | 61.64 | 58.00 | 39.58 | 37.50 | 49.18$_{\downarrow 3.20}$ |
| DOPA$_{-pro}$ | 55.81 | 59.36 | 38.41 | 37.21 | 47.70$_{\downarrow 1.70}$ | 61.89 | 58.31 | 39.21 | 44.60 | 51.00$_{\downarrow 1.38}$ |
| DOPA$_{uni}$ | 57.46 | 59.67 | 38.52 | 34.57 | 47.56$_{\downarrow 1.84}$ | 63.57 | 58.14 | 38.87 | 42.17 | 50.69$_{\downarrow 1.69}$ |
| DOPA | 59.29 | 59.87 | 39.15 | 39.29 | 49.40 | 64.93 | 59.22 | 39.92 | 45.44 | 52.38 |

Table 3: Ablation study results on `LLaMA3.2-3B` and `Qwen3-1.7B`.

may change the original entities. As an alternative, we compare with KNN and DrICL, which are not affected by the type of task.

For classification tasks in Table 1, DOPA shows noticeable performance disadvantages only in a few cases, underscoring its effectiveness in handling distribution-shifted scenarios. Moreover, Wilcoxon Signed-Rank Tests conducted across the 9 evaluation tasks indicate that DOPA significantly outperforms all baseline methods. In contrast, some of the latest baselines fail to consistently outperform random selection in OOD settings. For example, under `LLaMA3.2-3B`, the Random method frequently ranks second or third best, highlighting the persistent challenges of distribution shift. In such cases, relying solely on semantic retrieval (e.g., KNN) results in unstable performance. DrICL, which leverages LLM feedback to distinguish positive and negative samples and trains a dense retriever, generally outperforms KNN on average. Furthermore, the Rewrite approach proves less effective, as the strict unavailability of target domain samples limits the quality of rewritten prompts. Lastly, the influence-based retrieval method InfICL achieves comparable performance than DOPA in a few cases (e.g., `Qwen3-1.7B` and `GPT3.5-turbo` on TD), but remains unstable—performing worst on NLI with `GPT2-xl` and `LLaMA3.2-3B`, and on SA with `Gemma2-2B`.

For generative NER tasks in Table 2, we observe that DOPA yields greater performance improvements, which can be attributed to the higher difficulty of NER tasks compared to classification tasks, making them more susceptible to the distribution of demonstration samples. Besides, we find that KNN-based retrieval benefits lightweight LLMs that can be locally deployed, as these models rely more on external examples to guide their predictions. However, for larger models like `GPT4o-mini` and `GPT3.5-turbo`, KNN has a negative effect. This may be attributed to their stronger reasoning abilities and greater sensitivity to distribution shifts, making them more prone to being misled by semantically retrieved but distributionally mismatched examples. Building on the observed performance gains, we conduct the following analytical experiments to further investigate the underlying mechanisms of DOPA.

### 4.3 EXPERIMENTAL ANALYSIS

**Ablation Study**. To further validate the necessity of the key components in DOPA, we compare the following variants of DOPA to demonstrate the results of the ablation study. **DOPA$_{-pro}$** refers to a setting where no OOD proxy is used during demonstration retrieval, and sample representation similarity is solely relied upon for retrieval. **DOPA$_{-sim}$** indicates that no semantic similarity constraint is applied; instead, sample selection is performed directly based on the OOD proxy. **DOPA$_{-mah}$** indicates that the Mahalanobis distance–based diversity constraint is not applied. **DOPA$_{uni}$** indicates replacing the LLM-based target domain proxy with a uniform distribution to empirically validate Lemma 1. We report how the average performance across different tasks varies with different variants in Table 3. Overall, all variants lead to performance degradation, with the smallest drop observed in DOPA$_{-mah}$, followed by DOPA$_{-pro}$ and DOPA$_{uni}$, and the largest in DOPA$_{-sim}$. This highlights the positive contributions of each key component in DOPA: the OOD proxy is used for coarse filtering and selecting samples approximating the target domain, semantic similarity alignment further refines the retrieval, and the diversity constraint ensures the richness of the demonstration samples, where semantic similarity remains the most critical factor for retrieving relevant samples. Moreover, using a uniform proxy (DOPA$_{uni}$) leads to the second-largest performance drop, indicating that the LLM-based proxy is reasonable, which also supports the validity of Lemma 1. To sum up, incorporating proxy-based filtering and enforcing diversity constraints further enhance retrieval quality and model performance, underscoring the core contributions of DOPA.

**Exploration of $k$.** We conduct an exploration of the value of $k \in \{300, 500, 800, 1000\}$ to inves-

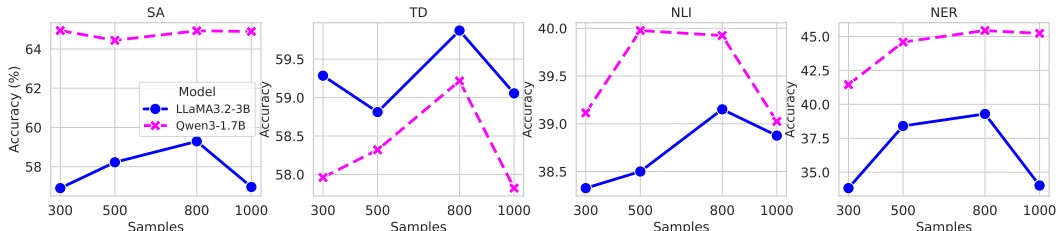

Figure 2: Performance influence of $k$ on `LLaMA3.2-3B` and `Qwen3-1.7B` across tasks.

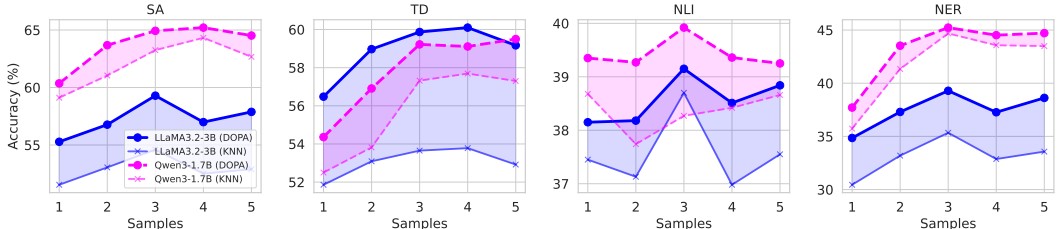

Figure 3: Performance influence of $N$ on DOPA and KNN based on `LLaMA3.2-3B` and `Qwen3-1.7B`, the shaded areas with corresponding colors indicate the performance differences.

tigate its impact on demonstration selection and model performance in Figure 2. The experimental results demonstrate that too small a value of $k$ may limit the diversity of examples and reduce the model's generalization ability. Conversely, larger values of $k$ increase the number of demonstrations but may introduce noise by including less relevant or redundant examples, potentially degrading model performance. Through systematic experiments across multiple tasks and datasets, we identify $k = 800$ as an optimal unified choice to balance the number of demonstrations across different tasks, even though $k = 800$ is not the optimal value in some cases. Fixing $k$ at a unified value simplifies the demonstration selection process, enhances consistency across tasks, and facilitates more stable and comparable model performance evaluation.

**Exploration of** $N$. We conduct an exploration of the value of $N \in \{1, 2, 3, 4, 5\}$ to investigate its impact on model performance in Figure 3. In addition, we select KNN as a baseline for comparison because it is compatible across different model tasks and yields stable results. Note that $N$ corresponds to a total of $N \times |Y|$ samples in demonstration. We observe a rising trend in performance as the number of demonstrations increases, with the model's performance gradually saturating as more demonstrations are added Min et al. (2022). Different models and tasks have varying demonstration requirements for performance saturation. For example, in the SA task, `Qwen3-1.7B` reaches peak performance at $N = 4$, while `LLaMA3.2-3B` peaks at $N = 3$. Regardless of the value of $N$, DOPA consistently achieves considerable performance improvements over KNN. In our implementation, for simplicity in the main experiments, we uniformly set $N = 3$.

**Visualization**. We verify the effectiveness of the OOD proxy by visualizing the proxy-based selection results. Specifically, we demonstrate the behavioral differences among $\mathcal{D}_S, \hat{\mathcal{D}}_S, \mathcal{D}_T$ by computing BERT-based energy scores Liu et al. (2020), and estimate their distributions using Kernel Density Estimation (KDE) Wkeglarczyk (2018). The reason we choose to fit the distribution of energy scores rather than use the commonly adopted t-SNE representation visualization is that differences in representations do not fully capture the OOD tendencies of samples. The visualization results in Figure 4 confirm that the proposed OOD proxy can select samples from the source domain that are closer to the target domain, as the distribution curve of the proxy in Figure 4a is more similar to and overlaps more with that of the target. In contrast, the representations distribution of the proxy-selected samples in Figure 4b is closer to the source domain, indicating that they still maintain a certain semantic distance from the target domain.

Additionally, to demonstrate the effectiveness of the diversity constraint, we select the first 1000 test samples and compute the Euclidean distances between the retrieved demonstrations and their corresponding test samples under both with MahDist and w/o MahDist settings. To better observe

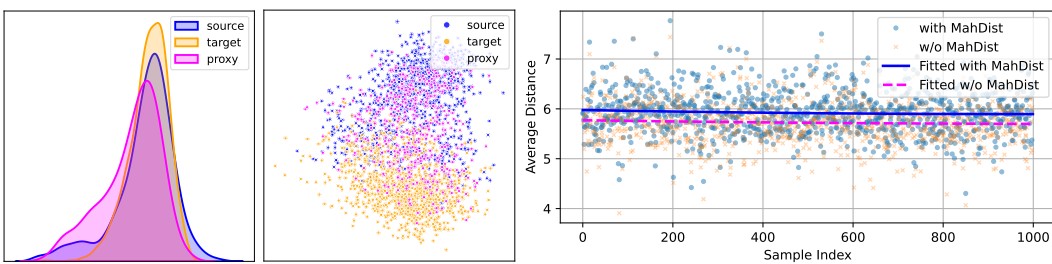

(a) KDE visualization results.

(b) t-SNE visualization results.

(c) Euclidean distance comparison to target domain samples for retrieval results with and without the diversity constraint (with MahDist and w/o MahDist).

Figure 4: Different visualization results on *sst*.

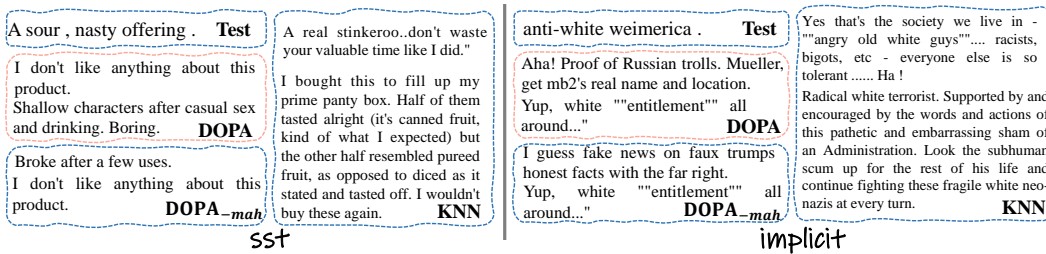

Figure 5: Case study on sst and implicit.

the overall distance characteristics, we also include corresponding fitted curves for visualization in Figure 4c. The fitted curve for w/o MahDist consistently lies below that of with MahDist, indicating that the diversity constraint indeed promotes more varied retrieval results. But this diversity is controlled—the with MahDist curve does not deviate significantly from w/o MahDist, suggesting that DOPA does not introduce excessive semantic drift.

In summary, the visualization results provide evidence for the effectiveness of DOPA from two perspectives: it helps retrieve demonstrations that exhibit similar behavior to target domain samples while maintaining high diversity, thereby enhancing the performance of ICL. We also observe similar trends across the remaining datasets. We include additional visualization results in Appendix D.

**Case Study**. The examples in the figure illustrate that both DOPA and DOPA$_{-mah}$ select samples with stylistic expressions closely aligned with the test inputs, capturing similar tone, sentence structure, and emotional/toxicity intensity. However, DOPA demonstrates slightly better diversity: in sst, while both methods retrieve strongly negative, concise opinions, DOPA's samples vary slightly more in content and phrasing. In the implicit task, both methods capture politically charged and provocative language, but DOPA avoids redundancy by selecting stylistically consistent yet semantically distinct sentences. In contrast, KNN selects samples that, although semantically related, deviate significantly in style—favoring longer or expository sentences that mismatch the terse nature of the test examples. Overall, DOPA achieves stronger style alignment with greater diversity, while KNN struggles to capture the nuanced stylistic cues of the target domain.

## 5 CONCLUSION

This paper demonstrates the effectiveness of OOD proxies in retrieving samples that closely resemble the target domain in ICL tasks with substantial distributional shifts. Building on this insight, we propose DOPA, a framework that operates without access to any additional target domain data, making it well-suited to real-world deployment constraints. To counteract the OOD proxies' undesirable bias toward short texts, DOPA incorporates a diversity constraint. Its effectiveness is validated across multiple widely used LLMs. In future work, we aim to extend our framework to a broader range of models and tasks, with a particular focus on developing more robust and effective proxy estimation methods when the target domain is unknown.

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

## A  THE USE OF LLMs

It should be noted that LLMs are involved in the translation and polishing of this manuscript. Furthermore, LLMs are utilized in the process of code development. However, we confirm that no instructions favoring LLMs in the review process have been added to the manuscript.

## B  THEORETICAL ANALYSIS AND PROOF

The following provides a detailed proof of the boundedness of proxy errors.

*Proof.* We use shorthand notation: let $P_t := P_{\text{target}}$, $P_s := P_{\text{source}}$, $P_t^p := P_{\text{target}}^{\text{proxy}}$, and $P_s^p := P_{\text{source}}^{\text{proxy}}$.

We aim to bound the log-likelihood ratio error:

$$\Delta(x) := \left| \log \frac{P_t(x)}{P_s(x)} - \log \frac{P_t^p(x)}{P_s^p(x)} \right|$$

Applying the triangle inequality:

$$\Delta(x) = |[\log P_t(x) - \log P_t^p(x)] - [\log P_s(x) - \log P_s^p(x)]| \leq \left| \log \frac{P_t(x)}{P_t^p(x)} \right| + \left| \log \frac{P_s(x)}{P_s^p(x)} \right|$$

We now upper bound each term. Then, from the definition of KL divergence:

$$D_{\text{KL}}(P_t \| P_t^p) = \sum_x P_t(x) \log \frac{P_t(x)}{P_t^p(x)} \leq \varepsilon_t$$

Now, suppose for some $x$ we have $P_t(x) \geq m_t$ and

$$\left| \log \frac{P_t(x)}{P_t^p(x)} \right| > \frac{\varepsilon_t}{m_t}$$

Then,

$$P_t(x) \cdot \left| \log \frac{P_t(x)}{P_t^p(x)} \right| > m_t \cdot \frac{\varepsilon_t}{m_t} = \varepsilon_t$$

This contradicts the assumption $D_{\text{KL}}(P_t \| P_t^p) \leq \varepsilon_t$. Therefore, for all $x$:

$$\left| \log \frac{P_t(x)}{P_t^p(x)} \right| \leq \frac{\varepsilon_t}{m_t}$$

Analogously, we obtain:

$$\left| \log \frac{P_s(x)}{P_s^p(x)} \right| \leq \frac{\varepsilon_s}{m_s}$$

Combining the two bounds:

$$\Delta(x) \leq \frac{\varepsilon_t}{m_t} + \frac{\varepsilon_s}{m_s},$$

the proof of the theorem is complete. $\square$

The theorem shows that if the KL-divergence between the true distribution and its proxy is sufficiently small, and the probability mass at each point is lower bounded, then the deviation in log-probability ratios is controllable in expectation. Therefore, a properly constructed proxy distribution yields bounded error in tasks such as density ratio estimation or scoring, which verifies the effectiveness and reliability of using proxies.

Moreover, some methods propose a general approach by replacing the target-domain proxy with a uniform distribution. However, this strong assumption may lead to suboptimal solutions. Accordingly, we introduce Lemma 1 to illustrate the limitations of using a uniform distribution.

*Proof of Lemma 1.* We consider the case where the proxy distribution for the target domain is chosen as the uniform distribution over the support $\mathcal{X}$:

$$P_t^p(x) = \frac{1}{|\mathcal{X}|} \quad \text{for all } x \in \mathcal{X}$$

From Theorem 1, the error in the log-likelihood ratio satisfies:

$$\left| \log \frac{P_t(x)}{P_s(x)} - \log \frac{P_t^p(x)}{P_s^p(x)} \right| \leq \left| \log \frac{P_t(x)}{P_t^p(x)} \right| + \left| \log \frac{P_s(x)}{P_s^p(x)} \right|$$

We now focus on bounding the first term with the uniform proxy:

$$A_t(x) := \left| \log \frac{P_t(x)}{P_t^p(x)} \right| = |\log (P_t(x) \cdot |\mathcal{X}|)| = |\log P_t(x) + \log |\mathcal{X}||$$

From the definition of KL divergence between $P_t$ and uniform distribution $U$:

$$D_{\mathrm{KL}}(P_t \| U) = \sum_x P_t(x) \log \frac{P_t(x)}{1/|\mathcal{X}|} = \sum_x P_t(x)[\log P_t(x) + \log |\mathcal{X}|] = \log |\mathcal{X}| - H(P_t)$$

where $H(P_t) := -\sum_x P_t(x) \log P_t(x)$ is the Shannon entropy of $P_t$.

Now suppose that $P_t(x) \geq m_t > 0$ for all $x$. Following the same logic as in the proof of Theorem 1, we know that if:

$$\left| \log \frac{P_t(x)}{P_t^p(x)} \right| > \frac{D_{\mathrm{KL}}(P_t \| U)}{m_t}$$

Then this point would contribute more than $D_{\mathrm{KL}}(P_t \| U)$ to the KL divergence, leading to a contradiction. Therefore, for all $x$:

$$\left| \log \frac{P_t(x)}{P_t^p(x)} \right| \leq \frac{D_{\mathrm{KL}}(P_t \| U)}{m_t} = \frac{\log |\mathcal{X}| - H(P_t)}{m_t}$$

Substituting into the total bound in Theorem 1, we obtain:

$$\left| \log \frac{P_t(x)}{P_s(x)} - \log \frac{P_t^p(x)}{P_s^p(x)} \right| \leq \frac{\log |\mathcal{X}| - H(P_t)}{m_t} + \frac{\varepsilon_s}{m_s}$$

This upper bound is typically looser than the one obtained when $P_t^p$ approximates $P_t$ well (i.e., KL divergence is small), since $\log |\mathcal{X}| - H(P_t)$ can be large when $P_t$ is sharply peaked.

$\square$

## C  EXPERIMENTAL DETAILS

### C.1  DATASET DETAILS

We focus on four core NLP tasks from BOSS Yuan et al. (2023), a benchmark suite specifically designed to evaluate the robustness of language models under OOD scenarios: Sentiment Analysis (SA), Toxic Detection (TD), Natural Language Inference (NLI), and Named Entity Recognition (NER). To balance the number of samples, we randomly select 3,000 training samples per class from the original in-distribution dataset for SA and NLI, and 5,000 training samples per class for TD. Accordingly, for testing, we randomly sample up to 1,000 instances per class from the target domain for SA and NLI, and 1,500 test samples per class for TD. For the NER task, we select 10,000 samples from source dataset that contain only "Location", "Organization", or "Person" entities to unify the label space and select all eligible samples from the target domain for testing. In our experiments, we do not use the conll dataset because it contains a large number of annotation errors, which could lead to unreliable and unmeasurable outcomes for model evaluation.

### C.2 Baseline Details

We provide a detailed introduction of the baseline methods used in this section.

- Random Peng et al. (2024). We randomly select the required number of samples from the source domain to construct demonstrations. To reduce performance variance caused by randomness, we repeat this process five times and report the average results for comparison.
- KNN Liu et al. (2022). We use the SimCSE representations of samples as the retrieval basis and construct demonstrations by selecting the top nearest samples to the test sample in the representation space.
- DrICL Luo et al. (2023). We first use KNN to select the top 30 candidate samples that are most similar to the test sample. These candidates are then ranked by quantifying their individual contributions to the LLM's actual predictions (We use LLaMA3.2-3B in LLMs that can not be deployed locally). The top 10 are treated as positive examples and the bottom 10 as negative ones to train a dual-encoder neural retriever, GTR Ni et al. (2022), which is subsequently used for demonstration retrieval.
- Rewrite Madine (2024). We perform KNN-based demonstration retrieval and rewrite the retrieved samples according to the style of the test sample, so that the demonstrations better align with the target domain. In contrast to the original method, we adapt the rewriting strategy under a strict target-unavailable setting, where only a single test instance is exposed at a time, rather than a set of target samples.
- InfICL S. et al. (2024). It estimates the influence of each candidate demonstration on the model's prediction for a given test input, and to select those demonstrations that have the most beneficial effect. By leveraging gradient-based influence approximations, the method identifies which demonstrations most positively affect the model's output distribution without requiring extensive evaluation over all combinations.

### C.3 More Experimental Settings

To prevent potential bias caused by an imbalanced number of samples per label in the demonstrations, we retrieve the same number of samples $N$ for each label. Therefore, for classification tasks, the total number of demonstrations is $N \times |Y|$, where $|Y|$ is the number of labels. However, for generative tasks that do not involve specific class labels, we directly set the number of demonstrations to $N$. For classification tasks, we set the number of demonstrations $C$ in the initial demonstration set to $|Y|$, while for generative tasks, we directly set $C$ to 1. For instruction fine-tuning, we use the source domain data and convert it into training samples following the instruction format of BOSS. During training, we apply LoRA with a learning rate of 1e-5 for one epoch. For `GPT4o-mini` and `GPT3.5-turbo`, we make the call using the interface provided by xi-ai[7].

To compare the performance of DOPA and baselines across multiple datasets, we employ the Wilcoxon Signed-Rank Test which is widely used for model comparison across multiple benchmarks Demsar (2006). This non-parametric statistical test is specifically designed for paired samples and does not assume normality of the underlying distribution. In our setting, the paired observations correspond to the performance scores of the two models (DOPA and any other baseline) on the same datasets. If DOPA shows statistically significant improvements ($p \leq 0.05$) over all baselines, we denote it as DOPA*.

## D More Visualization Results

We further present KDE distributions of sample representations across various tasks in Figure 6 to demonstrate the generality of DOPA in selecting appropriate samples. Overall, the samples selected by the proxy consistently exhibit a distribution that shifts away from the source domain and moves closer to the target domain. For example, on the *implicit_hate* dataset, the proxy-based distribution almost completely overlaps with that of the target domain. This demonstrates DOPA's capability to effectively identify samples with similar underlying distributions to the target domain. But we also observe that in a few cases (e.g., *anli*), the proxy-based distribution fails to effectively deviate

---

[7] https://api.xi-ai.cn/

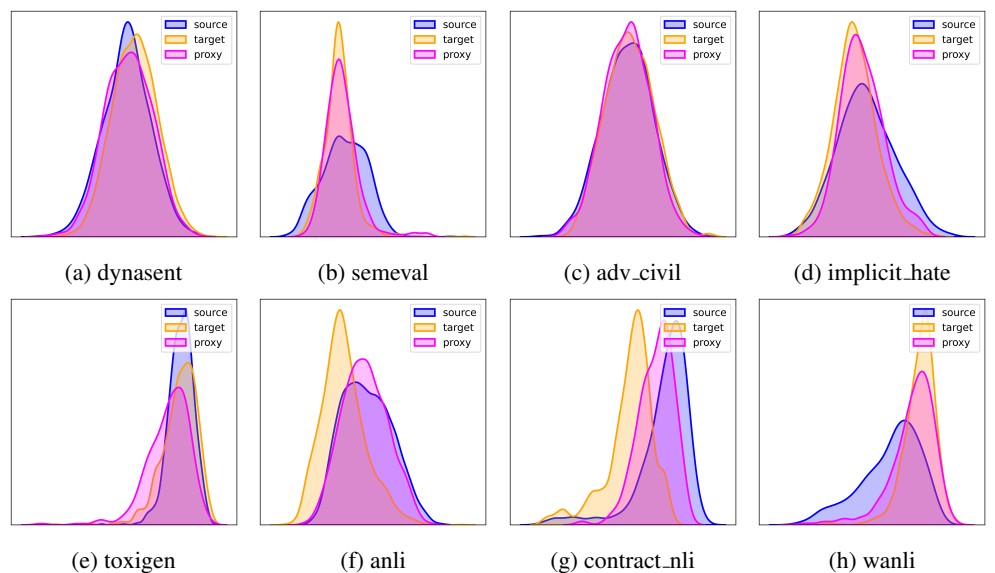

Figure 6: Different KDE visualization results on all classification tasks.

from the source domain. This may be attributed to the nature of *anli* itself, which is a human-crafted adversarial benchmark, making it challenging for the model to accurately capture its characteristics. We do not perform the corresponding visualization experiments on the NER dataset because it is not a sentence-level task, making it difficult to obtain the relevant probability distributions.

Figure 7 illustrates the effect of the diversity constraint across additional datasets. Similar to Figure 4c, the curve fitted under the with MahDist setting demonstrates greater diversity. Together with the ablation study, this provides strong evidence for the effectiveness of the diversity constraint component in DOPA.

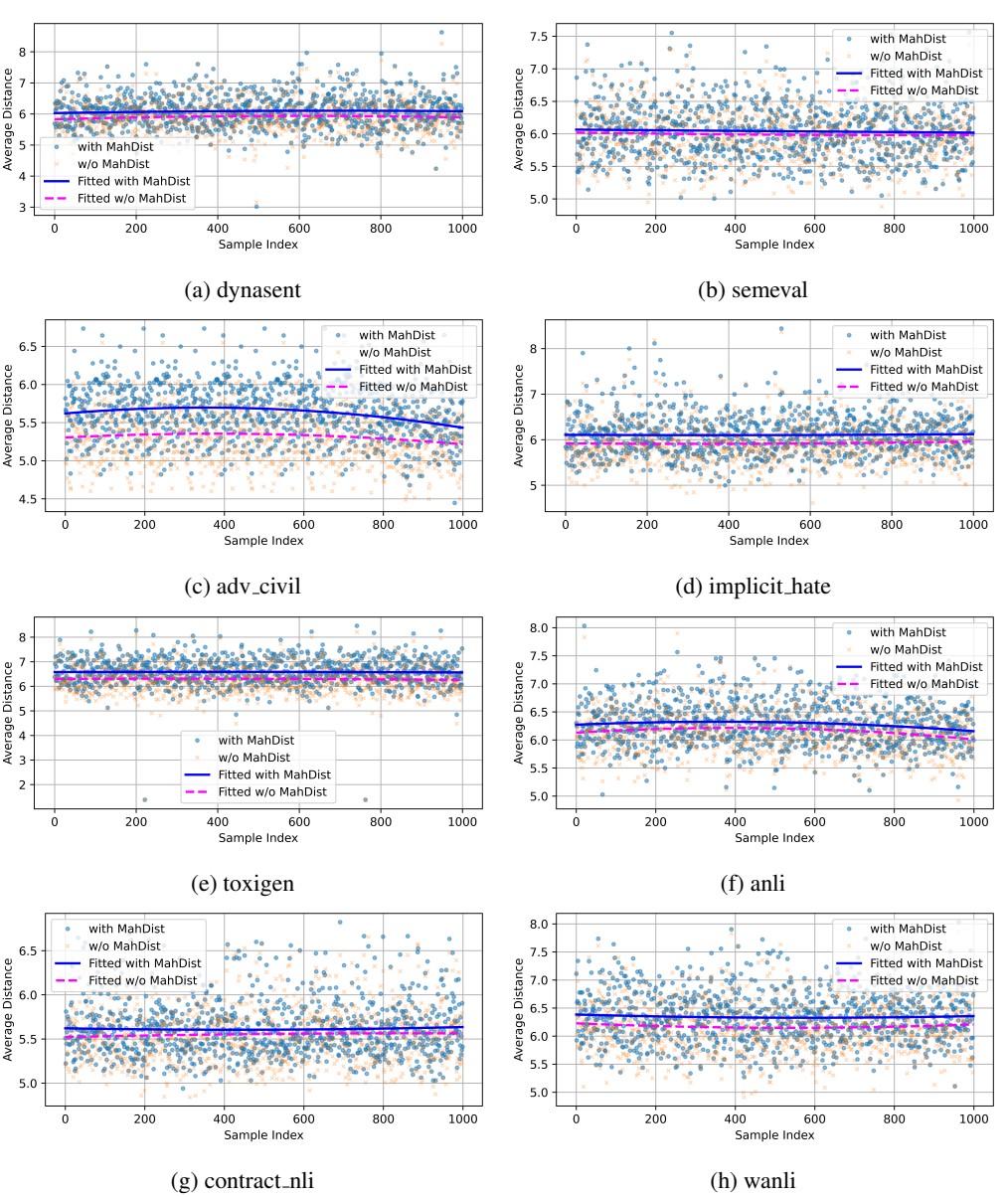

Figure 7: Euclidean distance comparison to target domain samples for retrieval results with and without the diversity constraint (with MahDist and w/o MahDist) on all tasks.

