# OpenReview forum: "Toward Robust In-Context Learning: Leveraging Out-of-distribution Proxies for Target Inaccessible Demonstration Retrieval"
_ICLR.cc/2026/Conference — ICLR 2026 Conference Withdrawn Submission_

### Official Review · Reviewer_6vej · 2025-10-29

**Soundness:** 1
**Presentation:** 1
**Contribution:** 2
**Rating:** 2
**Confidence:** 5

**Summary:**

The paper tackles ICL under distribution shift when the target domain is inaccessible and proposes DOPA. It scores source examples by a log-perplexity ratio between the two proxies to approximate a density-ratio–style OOD score, then performs multi-granularity retrieval. Experiments on the OOD benchmark show consistent gains over baselines.

**Strengths:**

- The “target-inaccessible” setup is realistic; using a pair of proxies is a coherent way to approximate density ratios without target data.


- Consistent improvements across tasks and models.

**Weaknesses:**

- A core appeal of ICL is improving performance without additional model training. DOPA introduces an instruction-tuned source proxy and relies on dual perplexity scoring, which shifts the method toward a trained, retrieval-time pipeline rather than pure in-context use. The paper should justify this trade-off and more clearly position DOPA relative to standard “no-finetune” ICL.
- The main components—proxy scoring akin to a density ratio, similarity-based ranking, and diversity control (Mahalanobis)—are each established (as cited in the paper). The paper should sharpen what is conceptually new and clearly delineate how its OOD-proxy estimation differs from closely related prior work [1], beyond implementation details.


- Please justify the realism of the assumptions in the theorem in OOD settings and clarify whether they are standard in prior work. While KL–ball assumptions are common in distributionally-robust testing/estimation, global pointwise density lower bounds are much stronger and may fail on high-dimensional supports or heavy-tailed data.
More importantly, the proof appears incorrect. From $D\_{KL}(P_t\Vert P_t^{\text{proxy}})=\sum_x P_t(x)\log\frac{P_t(x)}{P_t^{\text{proxy}}(x)}\le\varepsilon_t$, the manuscript infers a pointwise two-sided bound
 $\big|\log\frac{P_t(x)}{P_t^{\text{proxy}}(x)}\big|\le \varepsilon_t/m_t$
 by contradiction using $P_t(x)\ge m_t$. This step ignores that the summands $P_t(x)\log\frac{P_t(x)}{P_t^{\text{proxy}}(x)}$ can be negative; hence a large negative log-ratio at some (x) does not contradict a small average KL. As written, the key inequality does not follow, so the theorem’s final bound is not established.


[1]Zhang et al. Your finetuned large language model is already a powerful out-of-distribution detector.

**Questions:**

- Do you train the same model to serve as the source-domain proxy? If so, how is this handled for closed-source models such as GPT-4 and GPT-3.5, which cannot be fine-tuned directly? Please clarify how the proxy is implemented or approximated in these cases. Also, please provide the finetuning curves.


- How do you guarantee that instruction-tuning the source proxy uses only source-domain data (no leakage of target distribution or template artifacts)? Please describe safeguards, data splits, and checks.

---

### Official Review · Reviewer_1bSD · 2025-10-30

**Soundness:** 3
**Presentation:** 1
**Contribution:** 2
**Rating:** 2
**Confidence:** 4

**Summary:**

This paper introduces DOPA (Demonstration Optimization via Proxy Assessment), a framework designed to improve the robustness of Large Language Models under Out-of-Distribution conditions. The core idea is to calculate the OOD score by the divergence between models' outputs of seen and unseen given data. The similarity and diversity (Mahalanobis Distance) are considered to search for  demonstrations as well.

**Strengths:**

- Reasonable motivation and intuition of key modules.
- The core idea of OOD score calculation is model-agnostic, which could motivate similar methods in other OOD approaches and applications.
- The bound provided is pure and mathematically elegant.

**Weaknesses:**

- About DOPA's generalization and the model-agnostic method. The models have sparse coverage, encompassing multiple model sizes. However, each size uses different architectures, lacking an ablation about scales on a fixed series, especially those larger than 7B.
- The limitation of vaccum bound is not discussed. $m_{\cdot}$ is the lower bound larger than 0, and the divergence's upper bound $\epsilon$ is combined as $\epsilon/m_{\cdot}$, an upper bound of an upper-unlimited value. The cases do happen according to the confidence and perplexity vary that $m_{\cdot}$ is near zero.
- The table design, e.g., 1pt row spacing and over-simplified entities, makes it hard to read, even though there's so much space available.

**Questions:**

- In some cases, Random and KNN beat other latest methods. Are there any reasons behind? How about an ablation grid of scale on a fixed LLM series?
- Uniform proxy simplifies the analyses, but how about practically specific cases and general prior assumptions that make the bound tight? Especially when using proof by contradiction in the Appendix, the derivation of the conclusion appears remarkably straightforward. Is there any more quantitative upper bound?

---

### Official Review · Reviewer_hZX7 · 2025-11-01

**Soundness:** 2
**Presentation:** 1
**Contribution:** 2
**Rating:** 2
**Confidence:** 3

**Summary:**

The paper tackles robust in-context learning (ICL) under out-of-distribution (OOD) shift when the target domain is inaccessible. It proposes DOPA, a demonstration retrieval framework that uses an OOD proxy to approximate the target distribution and guide selection from a source pool. DOPA scores source samples with the proxy signal, then enforces a global diversity constraint via (Mahalanobis) distance to avoid redundancy and proxy-induced myopia. Experiments across multiple LLMs (e.g., LLaMA-3.2-3B, Qwen-1.7B) and tasks (SA, TD, NLI, NER) show consistent gains over KNN and several baselines. Sensitivity studies vary the candidate pool size k and demonstration count N, and visualizations (KDE/energy, t-SNE) plus a small case study suggest proxy-selected demos are behaviorally closer to the target while remaining diverse.

**Strengths:**

1 Addresses the realistic “target-inaccessible” regime, where many ICL retrieval methods quietly assume access to target data.
2 Broad evaluation: Multiple tasks and two model families; ablations on k and N; qualitative analyses supporting the proxy’s behavioral alignment and the value of diversity.

**Weaknesses:**

1 It’s not fully clear how the OOD proxy is obtained, calibrated, and validated without any target access.
2 The experimental part focuses on in-context learning, however, the compared baselines only involve several baselines that are not very recent. The more complex ICL methods should be considered to improve comprehensiveness.
3 Scope of tasks: All are NLU-style classification. It remains unclear whether benefits hold for generation (reasoning, QA, code)

**Questions:**

Do results transfer to generation tasks?

---

### Note · Authors · 2025-11-13

I have read and agree with the venue's withdrawal policy on behalf of myself and my co-authors.